# Training Deep Networks without Learning Rates Through Coin Betting

**Francesco Orabona**[*]
Department of Computer Science
Stony Brook University
Stony Brook, NY
francesco@orabona.com

**Tatiana Tommasi**[*]
Department of Computer, Control, and
Management Engineering
Sapienza, Rome University, Italy
tommasi@dis.uniroma1.it

## Abstract

Deep learning methods achieve state-of-the-art performance in many application scenarios. Yet, these methods require a significant amount of hyperparameters tuning in order to achieve the best results. In particular, tuning the learning rates in the stochastic optimization process is still one of the main bottlenecks. In this paper, we propose a new stochastic gradient descent procedure for deep networks that does not require any learning rate setting. Contrary to previous methods, we do not adapt the learning rates nor we make use of the assumed curvature of the objective function. Instead, we reduce the optimization process to a game of betting on a coin and propose a learning-rate-free optimal algorithm for this scenario. Theoretical convergence is proven for convex and quasi-convex functions and empirical evidence shows the advantage of our algorithm over popular stochastic gradient algorithms.

## 1 Introduction

In the last years deep learning has demonstrated a great success in a large number of fields and has attracted the attention of various research communities with the consequent development of multiple coding frameworks (e.g., Caffe [Jia et al., 2014], TensorFlow [Abadi et al., 2015]), the diffusion of blogs, online tutorials, books, and dedicated courses. Besides reaching out scientists with different backgrounds, the need of all these supportive tools originates also from the nature of deep learning: it is a methodology that involves many structural details as well as several hyperparameters whose importance has been growing with the recent trend of designing deeper and multi-branches networks. Some of the hyperparameters define the model itself (e.g., number of hidden layers, regularization coefficients, kernel size for convolutional layers), while others are related to the model training procedure. In both cases, hyperparameter tuning is a critical step to realize deep learning full potential and most of the knowledge in this area comes from living practice, years of experimentation, and, to some extent, mathematical justification [Bengio, 2012].

With respect to the optimization process, stochastic gradient descent (SGD) has proved itself to be a key component of the deep learning success, but its effectiveness strictly depends on the choice of the initial learning rate and learning rate schedule. This has primed a line of research on algorithms to reduce the hyperparameter dependence in SGD—see Section 2 for an overview on the related literature. However, all previous algorithms resort on adapting the learning rates, rather than removing them, or rely on assumptions on the shape of the objective function.

In this paper we aim at removing at least one of the hyperparameter of deep learning models. We leverage over recent advancements in the stochastic optimization literature to design a backprop-

---

[*]The authors contributed equally.

agation procedure that does not have a learning rate at all, yet it is as simple as the vanilla SGD. Specifically, we reduce the SGD problem to the game of betting on a coin (Section 4). In Section 5, we present a novel strategy to bet on a coin that extends previous ones in a data-dependent way, proving optimal convergence rate in the convex and quasi-convex setting (defined in Section 3). Furthermore, we propose a variant of our algorithm for deep networks (Section 6). Finally, we show how our algorithm outperforms popular optimization methods in the deep learning literature on a variety of architectures and benchmarks (Section 7).

## 2   Related Work

Stochastic gradient descent offers several challenges in terms of convergence speed. Hence, the topic of learning rate setting has been largely investigated.

Some of the existing solutions are based on the use of carefully tuned momentum terms [LeCun et al., 1998b, Sutskever et al., 2013, Kingma and Ba, 2015]. It has been demonstrated that these terms can speed-up the convergence for convex smooth functions [Nesterov, 1983]. Other strategies propose scale-invariant learning rate updates to deal with gradients whose magnitude changes in each layer of the network [Duchi et al., 2011, Tieleman and Hinton, 2012, Zeiler, 2012, Kingma and Ba, 2015]. Indeed, scale-invariance is a well-known important feature that has also received attention outside of the deep learning community [Ross et al., 2013, Orabona et al., 2015, Orabona and Pal, 2015]. Yet, both these approaches do not avoid the use of a learning rate.

A large family of algorithms exploit a second order approximation of the cost function to better capture its local geometry and avoid the manual choice of a learning rate. The step size is automatically adapted to the cost function with larger/shorter steps in case of shallow/steep curvature. Quasi-Newton methods [Wright and Nocedal, 1999] as well as the natural gradient method [Amari, 1998] belong to this family. Although effective in general, they have a spatial and computational complexity that is square in the number of parameters with respect to the first order methods, which makes the application of these approaches unfeasible in modern deep learning architectures. Hence, typically the required matrices are approximated with diagonal ones [LeCun et al., 1998b, Schaul et al., 2013]. Nevertheless, even assuming the use of the full information, it is currently unclear if the objective functions in deep learning have enough curvature to guarantee any gain.

There exists a line of work on unconstrained stochastic gradient descent without learning rates [Streeter and McMahan, 2012, Orabona, 2013, McMahan and Orabona, 2014, Orabona, 2014, Cutkosky and Boahen, 2016, 2017]. The latest advancement in this direction is the strategy of reducing stochastic subgradient descent to coin-betting, proposed by Orabona and Pal [2016]. However, their proposed betting strategy is worst-case with respect to the gradients received and cannot take advantage, for example, of sparse gradients.

## 3   Definitions

We now introduce the basic notions of convex analysis that are used in the paper—see, e.g., Bauschke and Combettes [2011]. We denote by $\|\cdot\|_1$ the 1-norm in $\mathbb{R}^d$. Let $f : \mathbb{R}^d \to \mathbb{R} \cup \{\pm\infty\}$, the *Fenchel conjugate* of $f$ is $f^* : \mathbb{R}^d \to \mathbb{R} \cup \{\pm\infty\}$ with $f^*(\boldsymbol{\theta}) = \sup_{\boldsymbol{x} \in \mathbb{R}^d} \boldsymbol{\theta}^\top \boldsymbol{x} - f(\boldsymbol{x})$.

A vector $\boldsymbol{x}$ is a *subgradient* of a convex function $f$ at $\boldsymbol{v}$ if $f(\boldsymbol{v}) - f(\boldsymbol{u}) \leq (\boldsymbol{v} - \boldsymbol{u})^\top \boldsymbol{x}$ for any $\boldsymbol{u}$ in the domain of $f$. The *differential set* of $f$ at $\boldsymbol{v}$, denoted by $\partial f(\boldsymbol{v})$, is the set of all the subgradients of $f$ at $\boldsymbol{v}$. If $f$ is also differentiable at $\boldsymbol{v}$, then $\partial f(\boldsymbol{v})$ contains a single vector, denoted by $\nabla f(\boldsymbol{v})$, which is the *gradient* of $f$ at $\boldsymbol{v}$.

We go beyond convexity using the definition of weak quasi-convexity in Hardt et al. [2016]. This definition is relevant for us because Hardt et al. [2016] proved that $\tau$-weakly-quasi-convex objective functions arise in the training of linear recurrent networks. A function $f : \mathbb{R}^d \to \mathbb{R}$ is $\tau$-*weakly-quasi-convex* over a domain $B \subseteq \mathbb{R}^d$ with respect to the global minimum $\boldsymbol{v}^*$ if there is a positive constant $\tau > 0$ such that for all $\boldsymbol{v} \in B$, $f(\boldsymbol{v}) - f(\boldsymbol{v}^*) \leq \tau(\boldsymbol{v} - \boldsymbol{v}^*)^\top \nabla f(\boldsymbol{v})$. From the definition, it follows that differentiable convex function are also 1-weakly-quasi-convex.

**Betting on a coin.**   We will reduce the stochastic subgradient descent procedure to betting on a number of coins. Hence, here we introduce the betting scenario and its notation. We consider a

gambler making repeated bets on the outcomes of adversarial coin flips. The gambler starts with initial money $\epsilon > 0$. In each round $t$, he bets on the outcome of a coin flip $g_t \in \{-1, 1\}$, where $+1$ denotes heads and $-1$ denotes tails. We do not make any assumption on how $g_t$ is generated.

The gambler can bet any amount on either heads or tails. However, he is not allowed to borrow any additional money. If he loses, he loses the betted amount; if he wins, he gets the betted amount back and, in addition to that, he gets the same amount as a reward. We encode the gambler's bet in round $t$ by a single number $w_t$. The sign of $w_t$ encodes whether he is betting on heads or tails. The absolute value encodes the betted amount. We define $\text{Wealth}_t$ as the gambler's wealth at the end of round $t$ and $\text{Reward}_t$ as the gambler's net reward (the difference of wealth and the initial money), that is

$$\text{Wealth}_t = \epsilon + \sum_{i=1}^{t} w_i g_i \qquad \text{and} \qquad \text{Reward}_t = \text{Wealth}_t - \epsilon = \sum_{i=1}^{t} w_i g_i . \qquad (1)$$

In the following, we will also refer to a bet with $\beta_t$, where $\beta_t$ is such that

$$w_t = \beta_t \, \text{Wealth}_{t-1} . \qquad (2)$$

The absolute value of $\beta_t$ is the *fraction* of the current wealth to bet and its sign encodes whether he is betting on heads or tails. The constraint that the gambler cannot borrow money implies that $\beta_t \in [-1, 1]$. We also slighlty generalize the problem by allowing the outcome of the coin flip $g_t$ to be any real number in $[-1, 1]$, that is a *continuous coin*; wealth and reward in (1) remain the same.

# 4 Subgradient Descent through Coin Betting

In this section, following Orabona and Pal [2016], we briefly explain how to reduce subgradient descent to the gambling scenario of betting on a coin.

Consider as an example the function $F(x) := |x - 10|$ and the optimization problem $\min_x F(x)$. This function does not have any curvature, in fact it is not even differentiable, thus no second order optimization algorithm could reliably be used on it. We set the outcome of the coin flip $g_t$ to be equal to the negative subgradient of $F$ in $w_t$, that is $g_t \in \partial[-F(w_t)]$, where we remind that $w_t$ is the amount of money we bet. Given our choice of $F(x)$, its negative subgradients are in $\{-1, 1\}$. In the first iteration we do not bet, hence $w_1 = 0$ and our initial money is \$1. Let's also assume that there exists a function $H(\cdot)$ such that our betting strategy will guarantee that the wealth after $T$ rounds will be at least $H(\sum_{t=1}^{T} g_t)$ for any arbitrary sequence $g_1, \cdots, g_T$.

We claim that the average of the bets, $\frac{1}{T} \sum_{t=1}^{T} w_t$, converges to the solution of our optimization problem and the rate depends on how good our betting strategy is. Let's see how.

Denoting by $x^*$ the minimizer of $F(x)$, we have that the following holds

$$F\left(\frac{1}{T} \sum_{t=1}^{T} w_t\right) - F(x^*) \leq \frac{1}{T} \sum_{t=1}^{T} F(w_t) - F(x^*) \leq \frac{1}{T} \sum_{t=1}^{T} g_t x^* - \frac{1}{T} \sum_{t=1}^{T} g_t w_t$$

$$\leq \frac{1}{T} + \frac{1}{T} \left( \sum_{t=1}^{T} g_t x^* - H\left( \sum_{t=1}^{T} g_t \right) \right) \leq \frac{1}{T} + \frac{1}{T} \max_v \, vx^* - H(v)$$

$$= \frac{H^*(x^*) + 1}{T},$$

where in the first inequality we used Jensen's inequality, in the second the definition of subgradients, in the third our assumption on $H$, and in the last equality the definition of Fenchel conjugate of $H$.

In words, we used a gambling algorithm to find the minimizer of a non-smooth objective function by accessing its subgradients. All we need is a good gambling strategy. Note that this is just a very simple one-dimensional example, but the outlined approach works in any dimension and for any convex objective function, even if we just have access to stochastic subgradients [Orabona and Pal, 2016]. In particular, if the gradients are bounded in a range, the same reduction works using a continuous coin.

Orabona and Pal [2016] showed that the simple betting strategy of $\beta_t = \frac{\sum_{i=1}^{t-1} g_i}{t}$ gives optimal growth rate of the wealth and optimal worst-case convergence rates. However, it is not data-dependent so it does not adapt to the sparsity of the gradients. In the next section, we will show an actual betting strategy that guarantees optimal convergence rate and adaptivity to the gradients.

---
**Algorithm 1** COntinuous COin Betting - COCOB
---
1: Input: $L_i > 0, i = 1, \cdots, d$; $\boldsymbol{w}_1 \in \mathbb{R}^d$ (initial parameters); $T$ (maximum number of iterations); $F$ (function to minimize)
2: Initialize: $G_{0,i} \leftarrow L_i$, $\text{Reward}_{0,i} \leftarrow 0$, $\theta_{0,i} \leftarrow 0$, $i = 1, \cdots, d$
3: **for** $t = 1, 2, \ldots, T$ **do**
4:     Get a (negative) stochastic subgradient $\boldsymbol{g}_t$ such that $\mathbb{E}[\boldsymbol{g}_t] \in \partial[-F(\boldsymbol{w}_t)]$
5:     **for** $i = 1, 2, \ldots, d$ **do**
6:         Update the sum of the absolute values of the subgradients: $G_{t,i} \leftarrow G_{t-1,i} + |g_{t,i}|$
7:         Update the reward: $\text{Reward}_{t,i} \leftarrow \text{Reward}_{t-1,i} + (w_{t,i} - w_{1,i})g_{t,i}$
8:         Update the sum of the gradients: $\theta_{t,i} \leftarrow \theta_{t-1,i} + g_{t,i}$
9:         Calculate the fraction to bet: $\beta_{t,i} = \frac{1}{L_i}\left(2\sigma\left(\frac{2\theta_{t,i}}{G_{t,i}+L_i}\right) - 1\right)$, where $\sigma(x) = \frac{1}{1+\exp(-x)}$
10:       Calculate the parameters: $w_{t+1,i} \leftarrow w_{1,i} + \beta_{t,i}\left(L_i + \text{Reward}_{t,i}\right)$
11:     **end for**
12: **end for**
13: Return $\bar{\boldsymbol{w}}_T = \frac{1}{T}\sum_{t=1}^{T}\boldsymbol{w}_t$ or $\boldsymbol{w}_I$ where $I$ is chosen uniformly between 1 and $T$
---

## 5 The COCOB Algorithm

We now introduce our novel algorithm for stochastic subgradient descent, COntinuous COin Betting (COCOB), summarized in Algorithm 1. COCOB generalizes the reasoning outlined in the previous section to the optimization of a function $F : \mathbb{R}^d \to \mathbb{R}$ with bounded subgradients, reducing the optimization to betting on $d$ coins.

Similarly to the construction in the previous section, the outcomes of the coins are linked to the stochastic gradients. In particular, each $g_{t,i} \in [-L_i, L_i]$ for $i = 1, \cdots, d$ is equal to the coordinate $i$ of the negative stochastic gradient $\boldsymbol{g}_t$ of $F$ in $\boldsymbol{w}_t$. With the notation of the algorithm, COCOB is based on the strategy to bet a signed fraction of the current wealth equal to $\frac{1}{L_i}\left(2\sigma\left(\frac{2\theta_{t,i}}{G_{t,i}+L_i}\right) - 1\right)$, where $\sigma(x) = \frac{1}{1+\exp(-x)}$ (lines 9 and 10). Intuitively, if $\frac{\theta_{t,i}}{G_{t,i}+L_i}$ is big in absolute value, it means that we received a sequence of equal outcomes, i.e., gradients, hence we should increase our bets, i.e., the absolute value of $w_{t,i}$. Note that this strategy assures that $|w_{t,i}g_{t,i}| < \text{Wealth}_{t-1,i}$, so the wealth of the gambler is always positive. Also, it is easy to verify that the algorithm is scale-free because multiplying all the subgradients and $L_i$ by any positive constant it would result in the same sequence of iterates $w_{t,i}$.

Note that the update in line 10 is carefully defined: The algorithm does not use the previous $w_{t,i}$ in the update. Indeed, this algorithm belongs to the family of the Dual Averaging algorithms, where the iterate is a function of the average of the past gradients [Nesterov, 2009].

Denoting by $\boldsymbol{w}^*$ a minimizer of $F$, COCOB satisfies the following convergence guarantee.

**Theorem 1.** *Let $F : \mathbb{R}^d \to \mathbb{R}$ be a $\tau$-weakly-quasi-convex function and assume that $\boldsymbol{g}_t$ satisfy $|g_{t,i}| \le L_i$. Then, running COCOB for $T$ iterations guarantees, with the notation in Algorithm 1,*

$$\mathbb{E}[F(\boldsymbol{w}_I)] - F(\boldsymbol{w}^*) \le \sum_{i=1}^{d} \frac{L_i + |w_i^* - w_{1,i}|\sqrt{\mathbb{E}\left[L_i(G_{T,i}+L_i)\ln\left(1 + \frac{(G_{T,i}+L_i)^2(w_i^* - w_{1,i})^2}{L_i^2}\right)\right]}}{\tau T},$$

*where the expectation is with respect to the noise in the subgradients and the choice of $I$. Moreover, if $F$ is convex, the same guarantee with $\tau = 1$ also holds for $\boldsymbol{w}_T$.*

The proof, in the Appendix, shows through induction that betting a fraction of money equal to $\beta_{t,i}$ in line 9 on the outcomes $g_{i,t}$, with an initial money of $L_i$, guarantees that the wealth after $T$ rounds is at least $L_i \exp\left(\frac{\theta_{T,i}^2}{2L_i(G_{T,i}+L_i)} - \frac{1}{2}\ln\frac{G_{T,i}}{L_i}\right)$. Then, as sketched in Section 4, it is enough to calculate the Fenchel conjugate of the wealth and use the standard construction for the per-coordinate updates [Streeter and McMahan, 2010]. We note in passing that the proof technique is also novel because the one introduced in Orabona and Pal [2016] does not allow data-dependent bounds.

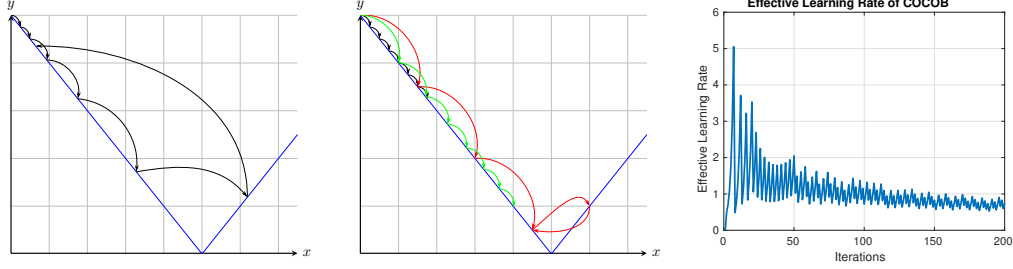

Figure 1: Behaviour of COCOB (left) and gradient descent with various learning rates and same number of steps (center) in minimizing the function $y = |x - 10|$. (right) The effective learning rates of COCOB. Figures best viewed in colors.

When $|g_{t,i}| = 1$, we have $\beta_{t,i} \approx \frac{\sum_{i=1}^{t-1} g_i}{t}$ that recovers the betting strategy in Orabona and Pal [2016]. In other words, we substitute the time variable with the data-dependent quantity $G_{t,i}$. In fact, our bound depends on the terms $G_{T,i}$ while the similar one in Orabona and Pal [2016] simply depends on $L_i T$. Hence, as in AdaGrad [Duchi et al., 2011], COCOB's bound is tighter because it takes advantage of sparse gradients.

COCOB converges at a rate of $\tilde{O}(\frac{\|\boldsymbol{w}^*\|_1}{\sqrt{T}})$ *without any learning rate to tune*. This has to be compared to the bound of AdaGrad that is[2] $O(\frac{1}{\sqrt{T}} \sum_{i=1}^{d} (\frac{(\boldsymbol{w}^*)^2}{\eta_i} + \eta_i))$, where $\eta_i$ are the initial learning rates for each coordinate. Usually all the $\eta_i$ are set to the same value, but from the bound we see that the optimal setting would require a different value for each of them. This effectively means that the optimal $\eta_i$ for AdaGrad are problem-dependent and typically unknown. Using the optimal $\eta_i$ would give us a convergence rate of $O(\frac{\|\boldsymbol{w}^*\|_1}{\sqrt{T}})$, that is exactly equal to our bound up to polylogarithmic terms. Indeed, the logarithmic term in the square root of our bound is the price to pay to be adaptive to any $\boldsymbol{w}^*$ and not tuning hyperparameters. This logarithmic term is unavoidable for any algorithm that wants to be adaptive to $\boldsymbol{w}^*$, hence our bound is optimal [Streeter and McMahan, 2012, Orabona, 2013].

To gain a better understanding on the differences between COCOB and other subgradient descent algorithms, it is helpful to compare their behaviour on the simple one-dimensional function $F(x) = |x - 10|$ already used in Section 4. In Figure 1 (left), COCOB starts from 0 and over time it increases in an exponential way the iterate $w_t$, until it meets a gradient of opposing sign. From the gambling perspective this is obvious: The wealth will increase exponentially because there is a sequence of identical outcomes, that in turn gives an increasing wealth and a sequence of increasing bets.

On the other hand, in Figure 1 (center), gradient descent shows a different behaviour depending on its learning rate. If the learning rate is constant and too small (black line) it will take a huge number of steps to reach the vicinity of the minimum. If the learning rate is constant and too large (red line), it will keep oscillating around the minimum, unless some form of averaging is used [Zhang, 2004]. If the learning rate decreases as $\frac{\eta}{\sqrt{t}}$, as in AdaGrad [Duchi et al., 2011], it will slow down over time, but depending of the choice of the initial learning rate $\eta$ it might take an arbitrary large number of steps to reach the minimum.

Also, notice that in this case the time to reach the vicinity of the minimum for gradient descent is not influenced in any way by momentum terms or learning rates that adapt to the norm of the past gradients, because the gradients are all the same. Same holds for second order methods: The function in figure lacks of any curvature, so these methods could not be used. Even approaches based on the reduction of the variance in the gradients, e.g. [Johnson and Zhang, 2013], do not give any advantage here because the subgradients are deterministic.

Figure 1 (right) shows the "effective learning" rate of COCOB that is $\tilde{\eta}_t := w_t \sqrt{\sum_{i=1}^{t} g_i^2}$. This is the learning rate we should use in AdaGrad to obtain the same behaviour of COCOB. We see a very

**Algorithm 2** COCOB-Backprop

---

1: Input: $\alpha > 0$ (default value = 100); $\boldsymbol{w}_1 \in \mathbb{R}^d$ (initial parameters); $T$ (maximum number of iterations); $F$ (function to minimize)
2: Initialize: $L_{0,i} \leftarrow 0, G_{0,i} \leftarrow 0, \text{Reward}_{0,i} \leftarrow 0, \theta_{0,i} \leftarrow 0, i = 1, \cdots,$ number of parameters
3: **for** $t = 1, 2, \ldots, T$ **do**
4:     Get a (negative) stochastic subgradient $\boldsymbol{g}_t$ such that $\mathbb{E}[\boldsymbol{g}_t] \in \partial[-F(\boldsymbol{w}_t)]$
5:     **for** each $i$-th parameter in the network **do**
6:         Update the maximum observed scale: $L_{t,i} \leftarrow \max(L_{t-1,i}, |g_{t,i}|)$
7:         Update the sum of the absolute values of the subgradients: $G_{t,i} \leftarrow G_{t-1,i} + |g_{t,i}|$
8:         Update the reward: $\text{Reward}_{t,i} \leftarrow \max(\text{Reward}_{t-1,i} + (w_{t,i} - w_{1,i})g_{t,i}, 0)$
9:         Update the sum of the gradients: $\theta_{t,i} \leftarrow \theta_{t-1,i} + g_{t,i}$
10:        Calculate the parameters: $w_{t,i} \leftarrow w_{1,i} + \frac{\theta_{t,i}}{L_{t,i}\max(G_{t,i}+L_{t,i},\alpha L_{t,i})}\left(L_{t,i} + \text{Reward}_{t,i}\right)$
11:    **end for**
12: **end for**
13: Return $\boldsymbol{w}_T$

---

interesting effect: The learning rate is not constant nor is monotonically increasing or decreasing. Rather, it is big when we are far from the optimum and small when close to it. However, we would like to stress that this behaviour has not been coded into the algorithm, rather it is a side-effect of having the optimal convergence rate.

We will show in Section 7 that this theoretical gain is confirmed in the empirical results.

## 6   Backprop and Coin Betting

The algorithm described in the previous section is guaranteed to converge at the optimal convergence rate for non-smooth functions and does not require a learning rate. However, it still needs to know the maximum range of the gradients on each coordinate. Note that for the effect of the vanishing gradients, each layer will have a different range of the gradients [Hochreiter, 1991]. Also, the weights of the network can grow over time, increasing the value of the gradients too. Hence, it would be impossible to know the range of each gradient beforehand and use any strategy based on betting.

By following the previous literature, e.g. [Kingma and Ba, 2015], we propose a variant of COCOB better suited to optimizing deep networks. We name it COCOB-Backprop and its pseudocode is in Algorithm 2. Although this version lacks the backing of a theoretical guarantee, it is still effective in practice as we will show experimentally in Section 7.

There are few differences between COCOB and COCOB-Backprop. First, we want to be adaptive to the maximum component-wise range of the gradients. Hence, in line 6 we constantly update the values $L_{t,i}$ for each variable. Next, since $L_{i,t-1}$ is not assured anymore to be an upper bound on $g_{t,i}$, we do not have any guarantee that the wealth $\text{Reward}_{t,i}$ is non-negative. Thus, we enforce the positivity of the reward in line 8 of Algorithm 2.

We also modify the fraction to bet in line 10 by removing the sigmoidal function because $2\sigma(2x)-1 \approx x$ for $x \in [-1, 1]$. This choice simplifies the code and always improves the results in our experiments. Moreover, we change the denominator of the fraction to bet such that it is at least $\alpha L_{t,i}$. This has the effect of restricting the value of the parameters in the first iterations of the algorithm. To better understand this change, consider that, for example, in AdaGrad and Adam with learning rate $\eta$ the first update is $w_{2,i} = w_{1,i} - \eta \text{SGN}(g_{1,i})$. Hence, $\eta$ should have a value smaller than $w_{1,i}$ in order to not "forget" the initial point too fast. In fact, the initialization is critical to obtain good results and moving too far away from it destroys the generalization ability of deep networks. Here, the first update becomes $w_{2,i} = w_{1,i} - \frac{1}{\alpha} \text{SGN}(g_{1,i})$, so $\frac{1}{\alpha}$ should also be small compared to $w_{1,i}$.

Finally, as in previous algorithms, we do not return the average or a random iterate, but just the last one (line 13 in Algorithm 2).

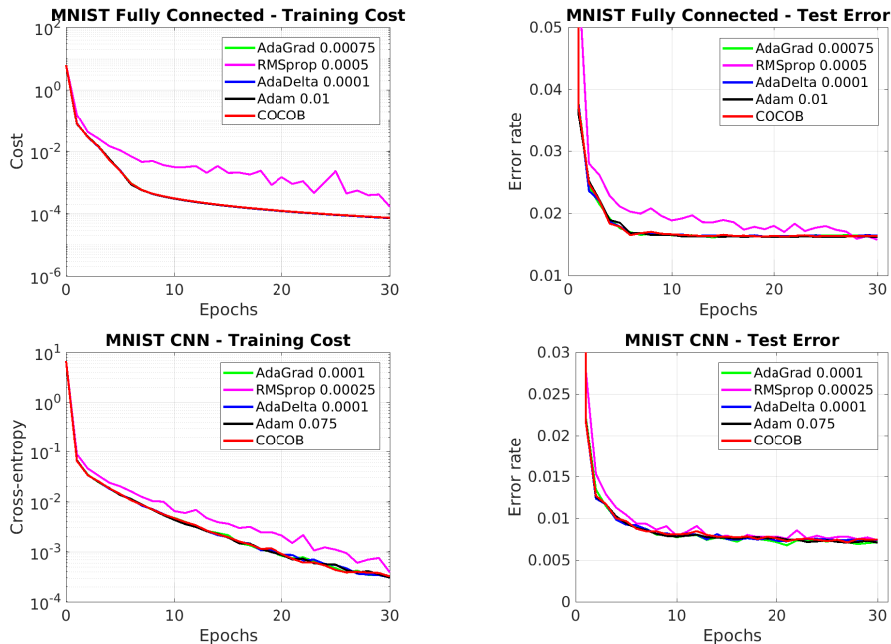

Figure 2: Training cost (cross-entropy) (left) and testing error rate (0/1 loss) (right) vs. the number epochs with two different architectures on MNIST, as indicated in the figure titles. The y-axis is logarithmic in the left plots. Figures best viewed in colors.

## 7 Empirical Results and Future Work

We run experiments on various datasets and architectures, comparing COCOB with some popular stochastic gradient learning algorithms: AdaGrad [Duchi et al., 2011], RMSProp [Tieleman and Hinton, 2012], Adadelta [Zeiler, 2012], and Adam [Kingma and Ba, 2015]. For all the algorithms, but COCOB, we select their learning rate as the one that gives the best training cost a posteriori using a very fine grid of values[3]. We implemented[4] COCOB (following Algorithm 2) in Tensorflow [Abadi et al., 2015] and we used the implementations of the other algorithms provided by this deep learning framework. The best value of the learning rate for each algorithm and experiment is reported in the legend.

We report both the training cost and the test error, but, as in previous work, e.g., [Kingma and Ba, 2015], we focus our empirical evaluation on the former. Indeed, given a large enough neural network it is always possible to overfit the training set, obtaining a very low performance on the test set. Hence, test errors do not only depends on the optimization algorithm.

**Digits Recognition.** As a first test, we tackle handwritten digits recognition using the MNIST dataset [LeCun et al., 1998a]. It contains $28 \times 28$ grayscale images with 60k training data, and 10k test samples. We consider two different architectures, a fully connected 2-layers network and a Convolutional Neural Network (CNN). In both cases we study different optimizers on the standard cross-entropy objective function to classify 10 digits. For the first network we reproduce the structure described in the multi-layer experiment of [Kingma and Ba, 2015]: it has two fully connected hidden layers with 1000 hidden units each and ReLU activations, with mini-batch size of 100. The weights are initialized with a centered truncated normal distribution and standard deviation 0.1, the same small value 0.1 is also used as initialization for the bias. The CNN architecture follows the Tensorflow tutorial [5]: two alternating stages of $5 \times 5$ convolutional filters and $2 \times 2$ max pooling are followed by a fully connected layer of 1024 rectified linear units (ReLU). To reduce overfitting, 50% dropout noise is used during training.

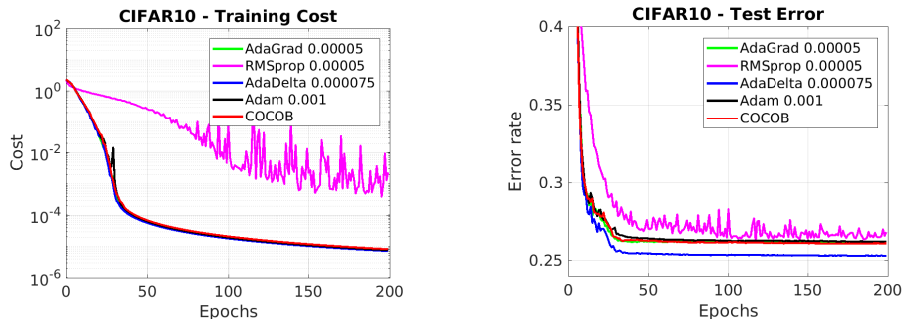

Figure 3: Training cost (cross-entropy) (left) and testing error rate (0/1 loss) (right) vs. the number epochs on CIFAR-10. The y-axis is logarithmic in the left plots. Figures best viewed in colors.

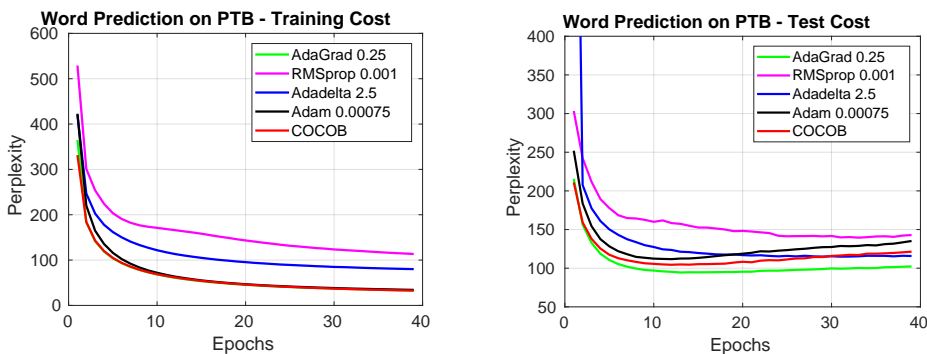

Figure 4: Training cost (left) and test cost (right) measured as average per-word perplexity vs. the number epochs on PTB word-level language modeling task. Figures best viewed in colors.

Training cost and test error rate as functions of the number of training epochs are reported in Figure 2. With both architectures, the training cost of COCOB decreases at the same rate of the best tuned competitor algorithms. The training performance of COCOB is also reflected in its associated test error which appears better or on par with the other algorithms.

**Object Classification.** We use the popular CIFAR-10 dataset [Krizhevsky, 2009] to classify $32 \times 32$ RGB images across 10 object categories. The dataset has 60k images in total, split into a training/test set of 50k/10k samples. For this task we used the network defined in the Tensorflow CNN tutorial[6]. It starts with two convolutional layers with 64 kernels of dimension $5 \times 5 \times 3$, each followed by a $3 \times 3 \times 3$ max pooling with stride of 2 and by local response normalization as in Krizhevsky et al. [2012]. Two more fully connected layers respectively of 384 and 192 rectified linear units complete the architecture that ends with a standard softmax cross-entropy classifier. We use a batch size of 128 and the input images are simply pre-processed by whitening. Differently from the Tensorflow tutorial, we do not apply image random distortion for data augmentation.

The obtained results are shown in Figure 3. Here, with respect to the training cost, our learning-rate-free COCOB performs on par with the best competitors. For all the algorithms, there is a good correlation between the test performance and the training cost. COCOB and its best competitor AdaDelta show similar classification results that differ on average $\sim 0.008$ in error rate.

**Word-level Prediction with RNN.** Here we train a Recurrent Neural Network (RNN) on a language modeling task. Specifically, we conduct word-level prediction experiments on the Penn Tree Bank (PTB) dataset [Marcus et al., 1993] using the 929k training words and its 73k validation words. We adopted the medium LSTM [Hochreiter and Schmidhuber, 1997] network architecture described in Zaremba et al. [2014]: it has 2 layers with 650 units per layer and parameters initialized uniformly in $[-0.05, 0.05]$, a dropout of 50% is applied on the non-recurrent connections, and the norm of the gradients (normalized by mini-batch size = 20) is clipped at 5.

We show the obtained results in terms of average per-word perplexity in Figure 4. In this task COCOB performs as well as Adagrad and Adam with respect to the training cost and much better than the other algorithms. In terms of test performance, COCOB, Adam, and AdaGrad all show an overfit behaviour indicated by the perplexity which slowly grows after having reached its minimum. Adagrad is the least affected by this issue and presents the best results, followed by COCOB which outperforms all the other methods. We stress again that the test performance does not depend only on the optimization algorithm used in training and that early stopping may mitigate the overfitting effect.

**Summary of the Empirical Evaluation and Future Work.** Overall, COCOB has a training performance that is on-par or better than state-of-the-art algorithms with perfectly tuned learning rates. The test error appears to depends on other factors too, with equal training errors corresponding to different test errors.

We would also like to stress that in these experiments, contrary to some of the previous reported empirical results on similar datasets and networks, the difference between the competitor algorithms is minimal or not existent when they are tuned on a very fine grid of learning rate values. Indeed, the very similar performance of these methods seems to indicate that all the algorithms are inherently doing the same thing, despite their different internal structures and motivations. Future more detailed empirical results will focus on unveiling what is the common structure of these algorithms that give rise to this behavior.

In the future, we also plan to extend the theory of COCOB beyond $\tau$-weakly-quasi-convex functions, characterizing the non-convexity present in deep networks. Also, it would be interesting to evaluate a possible integration of the betting framework with second-order methods.

## Acknowledgments

The authors thank the Stony Brook Research Computing and Cyberinfrastructure, and the Institute for Advanced Computational Science at Stony Brook University for access to the high-performance SeaWulf computing system, which was made possible by a $1.4M National Science Foundation grant (#1531492). The authors also thank Akshay Verma for the help with the TensorFlow implementation and Matej Kristan for reporting a bug in the pseudocode in the previous version of the paper. T.T. was supported by the ERC grant 637076 - RoboExNovo. F.O. is partly supported by a Google Research Award.

## Footnotes

[2]The AdaGrad variant used in deep learning does not have a convergence guarantee, because no projections are used. Hence, we report the oracle bound in the case that projections are used inside the hypercube with dimensions $|w_i^*|$.

[3] [0.00001, 0.000025, 0.00005, 0.000075, 0.0001, 0.00025, 0.0005, 0.00075, 0.001, 0.0025, 0.005, 0.0075, 0.01, 0.02, 0.05, 0.075, 0.1]

[4] `https://github.com/bremen79/cocob`

[5] `https://www.tensorflow.org/get_started/mnist/pros`

[6]`https://www.tensorflow.org/tutorials/deep_cnn`

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
