[Supplementary Material]

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

## A Proof of Theorem 1

First we state some technical lemmas that will be used in the proof of the convergence rate of COCOB.

**Lemma 1.** *[Orabona and Pal, 2016, extended version, Lemma 18] Define $f(x) = \beta \exp \frac{x^2}{2\alpha}$, for $\alpha, \beta > 0$. Then*

$$f^*(y) \leq |y|\sqrt{\alpha \log \left( \frac{\alpha y^2}{\beta^2} + 1 \right)} - \beta .$$

**Lemma 2.** *Let $a \geq 2$. Then, with the notation of Algorithm 1, for any $t$ we have*

$$(1 + \beta_{t,i} g_{t,i}) \exp \left( \frac{\theta_{t-1,i}^2}{aL_i(G_{t-1,i} + L_i)} - \sum_{j=1}^{t-1} \frac{|g_{j,i}|}{a(G_{j-1,i} + L_i)} \right)$$

$$\geq \exp \left( \frac{\theta_{t,i}^2}{aL_i(G_{t,i} + L_i)} - \sum_{j=1}^{t} \frac{|g_{j,i}|}{a(G_{j-1,i} + L_i)} \right) .$$

*Proof.* The statement to prove is equivalent to

$$\ln(1 + \beta_t g_t) + \frac{\theta_{t-1}^2}{aL(L + \sum_{j=1}^{t-1} |g_j|)} \geq \frac{(\theta_t + g_t)^2}{aL(L + \sum_{j=1}^{t} |g_j|)} - \frac{|g_t|}{a(L + \sum_{l=1}^{t-1} |g_l|)},$$

where for clarity we dropped the index $i$.

Consider the function

$$\phi(x) = -\log(1 + \beta_t x) + \frac{(\theta_{t-1} + x)^2}{aL(G_{t-1} + |x|)} .$$

We have that $\phi(x)$ is piece-wise convex on $[-\infty, 0]$ and $[0, \infty]$. Hence, we have that

$$\phi(x) \leq \phi(0) + \frac{x}{L}(\phi(L) - \phi(0)), \forall 0 \leq x \leq L$$

$$\phi(x) \leq \phi(0) + \frac{x}{L}(\phi(0) - \phi(-L)), \forall -L \leq x \leq 0 .$$

Also, $\beta_t$ is such that $\phi(L) = \phi(-L)$. Hence, we have

$$\phi(x) \le \phi(0) + \frac{|x|}{L}(\phi(L) - \phi(0)), \forall -L \le x \le L,$$

that is

$$\frac{\theta_{t-1}^2}{aLG_{t-1}} - \frac{(\theta_{t-1} + g_t)^2}{aL(G_{t-1} + |g_t|)} + \log(1 + \beta_t g_t)$$
$$= \phi(0) - \phi(g_t)$$
$$\ge \frac{|g_t|}{L}(\phi(0) - \phi(L))$$
$$= \frac{|g_t|}{L}\left(\frac{\theta_{t-1}^2}{aLG_{t-1}} - \frac{(\theta_{t-1} + L)^2}{aL(G_{t-1} + L)} + \log(1 + \beta_t L)\right), \forall -L \le g_t \le L.$$

Using this relation we have that

$$\frac{\theta_{t-1}^2}{aLG_{t-1}} - \frac{(\theta_{t-1} + g_t)^2}{aL(G_{t-1} + |g_t|)} + \log(1 + \beta_t g_t)$$
$$\ge \frac{|g_t|}{L}\left(\frac{\theta_{t-1}^2}{aLG_{t-1}} - \frac{(\theta_{t-1} + L)^2}{aL_i(G_{t-1} + L)} + \log(1 + L\beta_t)\right)$$
$$= \frac{|g_t|}{L}\left(\frac{(G_{t-1} + L)\theta_{t-1}^2 - (\theta_{t-1}^2 + 2L\theta_{t-1})G_{t-1}}{aLG_{t-1}(G_{t-1} + L)} + \log(1 + L\beta_t)\right) - \frac{|g_t|}{a(G_{t-1} + L)}$$
$$= \frac{|g_t|}{L}\left(\frac{\theta_{t-1}^2}{aG_{t-1}(G_{t-1} + L)} - \frac{2\theta_{t-1}}{a(G_{t-1} + L)} + \log(1 + L\beta_t)\right) - \frac{|g_t|}{a(G_{t-1} + L)}.$$

We now use the Taylor expansion, to obtain

$$\log\left(1 + \frac{\exp(x) - 1}{\exp(x) + 1}\right) \ge \frac{x}{2} - \frac{x^2}{8} \qquad \forall x \in \mathbb{R}$$

and, using the expression of $\beta_t$, have

$$\log(1 + L\beta_t) = \log\left(1 + \frac{\exp\left(\frac{4\theta_{t-1}}{a(G_{t-1} + L)}\right) - 1}{\exp\left(\frac{4\theta_{t-1}}{a(G_{t-1} + L)}\right) + 1}\right) \ge \frac{2\theta_{t-1}}{a(G_{t-1} + L)} - \frac{2\theta_{t-1}^2}{a^2(G_{t-1} + L)^2}.$$

Hence the expression

$$\frac{\theta_{t-1}^2}{aG_{t-1}(G_{t-1} + L)} - \frac{2\theta_{t-1}}{a(G_{t-1} + L)} + \log(1 + L\beta_t) \ge \frac{\theta_{t-1}^2}{aG_{t-1}(G_{t-1} + L_i)} - \frac{2\theta_{t-1}^2}{a^2(G_{t-1} + L)^2}$$
$$\ge \frac{aL_i\theta_{t-1}^2 + aG_{t-1}\theta_{t-1}^2 - 2G_{t-1}\theta_{t-1}^2}{a^2G_{t-1}(G_{t-1} + L)^2}$$

is greater than zero if $a \ge 2$, that is true by definition of $a$. $\qquad\square$

We can now prove Theorem 1.

*Proof of Theorem 1.* First, assume that $\boldsymbol{w}_1 = \boldsymbol{0}$, then we will show how to remove this assumption.

Define $H_{t,i}(x) = L_i \exp\left(\frac{x^2}{2L_i(G_{t,i} + L_i)} - \sum_{j=1}^{t} \frac{|g_j|}{2(L_i + G_{j-1,i})}\right)$. By induction, we first prove that $\text{Wealth}_{t,i} \ge H_{t,i}(\theta_{t,i})$. For $t = 0$, it is obvious because $\text{Wealth}_{0,i} = L_i$. We now assume that $\text{Wealth}_{t-1} \ge H_{t-1,i}(\theta_{t-1,i})$. Note that $|\beta_{t,i} g_{t,i}| < 1$. Hence, using Lemma 2, we have

$$\begin{aligned}
\text{Wealth}_{t,i} = \text{Wealth}_{t-1,i} + g_{t,i} w_{t,i} &= \text{Wealth}_{t-1,i}(1 + g_{t,i}\beta_{t,i}) \\
&\ge (1 + g_{t,1}\beta_{t,i})H_{t-1,i}(\theta_{t-1,i}) \qquad (3) \\
&\ge H_{t,i}(\theta_{t,i}),
\end{aligned}$$

that proves the induction.

Now, in the convex case, using the fact that the stochastic subgradient are unbiased, the definition of the subgradients, and Jensen's inequality, we have

$$T\left(\mathbb{E}[F(\bar{\boldsymbol{w}}_T)] - F(\boldsymbol{w}^*)\right) \leq \sum_{t=1}^{T}(\mathbb{E}[F(\boldsymbol{w}_t)] - F(\boldsymbol{w}^*)) \leq \sum_{t=1}^{T}\mathbb{E}[(\boldsymbol{w}^* - \boldsymbol{w}_t)^\top \boldsymbol{g}_t]\,.$$

While, in the the $\tau$-weakly-quasi-convex case, we have

$$T\left(\mathbb{E}[F(\boldsymbol{w}_I)] - F(\boldsymbol{w}^*)\right) = \sum_{t=1}^{T}\left(\mathbb{E}[F(\boldsymbol{w}_t)] - F(\boldsymbol{w}^*)\right) \leq \tau \sum_{t=1}^{T}\mathbb{E}[(\boldsymbol{w}^* - \boldsymbol{w}_t)^\top \boldsymbol{g}_t]$$

by fact that the gradient are unbiased, the definition of $\boldsymbol{w}_I$, and the definition of $\tau$-weakly-quasi-convexity. Hence, the two cases are the same up to the factor $\tau$. We can then proceed in both cases with

$$\sum_{t=1}^{T}\mathbb{E}[(\boldsymbol{w}^* - \boldsymbol{w}_t)^\top \boldsymbol{g}_t] = \sum_{i=1}^{d}\sum_{t=1}^{T}\mathbb{E}[w_i^* g_{t,i} - g_{t,i} w_{t,i}] = \sum_{i=1}^{d}\mathbb{E}[L_i + w_i^* \theta_{T,i} - \text{Wealth}_{T,i}]\,. \quad (4)$$

Using the definition of Fenchel's conjugate, (4) and the lower bound on the wealth in (3), we have

$$\sum_{t=1}^{T}\mathbb{E}[(\boldsymbol{w}^* - \boldsymbol{w}_t)^\top \boldsymbol{g}_t] = \mathbb{E}\left[\sum_{i=1}^{d}(L_i + w_i^* \theta_{T,i} - \text{Wealth}_{T,i})\right]$$

$$\leq \mathbb{E}\left[\sum_{i=1}^{d}(L_i + w_i^* \theta_{T,i} - H_{T,i}(\theta_{T,i}))\right]$$

$$\leq \mathbb{E}\left[\sum_{i=1}^{d}L_i + \max_x\left(w_i^* x - H_{T,i}(x)\right)\right] = \mathbb{E}\left[\sum_{i=1}^{d}L_i + H_{T,i}^*(w_i^*)\right]\,.$$

Also, the concavity of the logarithm implies that $\frac{a-b}{a} \leq \ln a - \ln b$ for all $a \geq b > 0$. Hence

$$\sum_{j=1}^{T}\frac{|g_j|}{L_i + G_{j-1,i}} \leq \sum_{j=1}^{T}\frac{|g_j|}{G_{j,i}} \leq \sum_{j=1}^{T}(\ln G_{j,i} - \ln G_{j-1,i}) = \ln \frac{G_{T,i}}{G_{0,i}} = \ln \frac{G_{T,i}}{L_i}\,. \quad (5)$$

Using Lemma 1, the inequality in (5), and overapproximating, we have

$$H_{T,i}^*(w_i^*) \leq |w_i^*|\sqrt{L_i(G_{T,i} + L_i)\ln\left(1 + \frac{(G_{T,i} + L_i)^2 (w_i^*)^2}{L_i^2}\right)}\,.$$

Putting all together, using Jensen's inequality to bring the expectation under the square root, and dividing by $T$ give us the stated bound, with $\boldsymbol{w}_1 = \boldsymbol{0}$.

Now, running the algorithm on the function $\tilde{F}(\boldsymbol{w}) = F(\boldsymbol{w}_t + \boldsymbol{w}_1)$, for an arbitrary $\boldsymbol{w}_1$, would result in the update in Algorithm 1 and would guarantee the same upper bound on $\mathbb{E}[\tilde{F}(\bar{\boldsymbol{w}}_T)] - \tilde{F}(\boldsymbol{w}^*)$ that implies the stated bound. $\qquad\square$