[Reviews · NeurIPS 2017]

Reviewer 1



This paper extends work by Orabona and Pal on coin-flipping for optimal learning rate dynamics. I like the idea of a betting framework to tune learning rates over the course of training. My comments are as follows: 1) The abstract sounds as if the main contribution is the idea of coin-flipping for automatic learning rate tuning. However, this idea was introduced before as mentioned in the main text. What I am missing is a clear description (and quantitative demonstration) of the advantages of the proposed algorithm compared to previous work. This should also be more clearly stated in the abstract. 2) There is no quantitative comparison with the algorithm proposed in Orabona and Pal. 3) Do Orabona and Pal need the exact Lipschitz constant or just some bound? In the latter case their algorithm would also be applicable to deep neural networks. 4) I did not really get the intuition behind the coin flipping. In section 4 you introduce a very simple example |x - 10|. But here we only ever evaluate F(w_t), so we are never making a real step? In a learning scenario there would be an x_t that tracks the current state, and then w_t would be the learning rate? I am fairly confused, and I think it would be helpful to write one paragraph in which you walk through the first five steps of the algorithm. 5) Why is the performance of Adam so bumpy in Figure 2/3? I have a lot of experience with this optimiser and I have never seen such curves (unless the learning rates are chosen close to the upper limit). I’d suggest some tuning of the learning rates of all optimisation methods, otherwise the results are very hard to compare.

Reviewer 2



Summary: This paper is based on the notion (established in existing works) that subgradient descent can be interpreted as betting on a coin. It generalizes existing results to data-dependent bets and, consequently, data-dependent convergence bounds that improve upon the existing ones. The algorithm is then adapted for the training of neural networks and evaluated on this task experimentally. Quality: The derivations are mathematically sound. I verified the proof of Theorem 1. The changes made to adapt COCOB for the training of neural networks (Algo 1 --> Algo 2) are sensible and intuitive. However, the experimental evaluation of COCOB as a deep learning optimizer is insufficient in my opinion. The lack of theoretical guarantees for DL optimization makes a thorough experimental evaluation on a range of realistic problems indispensable. Small CNNs/MLPs on MNIST/CIFAR-10 are weak test problems. Furthermore, I don't agree with the premise of the comparison to other methods. It is stated that "default hyperparameters" are used for the competing methods. I don't think that something like a default learning rate really exists for Adam/RMSprop/etc. Even though COCOB does not require a learning rate, I think it is a somewhat unfair assessment to just compare against Adam/RMSprop/etc with a single learning rate (that might be considered "default" by some arbitrary standards). It would be much more convincing to compare against multiple learning rates to judge the performance of the learning-rate-free COCOB algorithm in light of the learning rate sensitivity of other methods. Clarity: The paper is well-written. All relevant concepts/definitions are properly introduced. Section 4 is very helpful in that it gently introduces the reader to the interpretation of subgradient descent as coin betting. To clarify the contribution of this paper, the authors could do a better job in explaining how their algorithm (and its convergence guarantee) is different from that in [Orabona & Pal, 2016]. Originality: This paper adopts the "coin-betting" paradigm of [Orabona & Pal, 2016] but presents improved theoretical results. At the same time, a variant of their algorithm is proposed as a practical method for deep learning optimization, which is a very novel, original approach compared to existing methods like Adam/RMSpop/AdaDelta/etc. Significance: This paper proposes a hyperparameter-free optimization algorithm for deep learning, which is a highly significant research area with a big potential impact on many deep learning researchers and practitioners. Final Evaluation: As explained above, I have my reservations about the experimental evaluation. Given the sizeable theoretical contribution of the paper, the novelty of the approach (in the deep learning context), and the big potential impact, I argue for acceptance in spite of that. Proposed Improvements: Add a thorough experimental evaluation on realistic deep learning optimization problems. For the competing methods, report results for various different learning rates. Minor Comments / Typos: - Typos * Line 32: should be "at least one of the hyperparameters of deep learning models" * Line 167: should be "a better understanding of the differences" - Line 103: F(x) is not differentiable at x=10 - Appendix Eq. (3), second line, should be g_{t, i} instead of g_{t, 1} - Appendix Eq. (5): there a some subscripts i missing in the g's

Reviewer 3



This paper presents an optimization strategy using coin betting, and a variant which works well for training neural networks. The optimizer is tested and compared to various stochastic optimization routines on some simple problems. I like this paper because it's an unusual optimization method which surprisingly seems to work reasonably well. It also has fewer tunable parameters than existing stochastic optimizers, which is nice. However, I'm giving it a marginal accept for the following reasons: - The experiments aren't very convincing. They're training some very simple models on some very small datasets, and results in this regime do not necessarily translate over to "real" problems/models. Further, success on these problems mostly revolves around preventing overfitting. I'd actually suggest additional experiments in _both_ directions - additional simple unit test-like experiments (like the |x - 10| experiment, but more involved - see e.g. the "unit tests for stochastic optimization" paper), and experiments on realistically large models on realistically large datasets. - I don't 100% buy the "without learning rates" premise, for the following reason: If you divide the right-most term of line 10 of Algorithm 2 by L_{t, i}, then the denominator simplifies to max(G_{t, i}/L_{t, i} + 1, \alpha). So, provided that \alpha is bigger than G_{t, i}/L_{t, i}, the updates effectively are getting scaled by \alpha. In general I would expect G_{t, i}/L_{t, i} to be smaller than \alpha = 100 particularly at the beginning of training, and as a result I'd expect the setting of \alpha to have a considerable effect on training. Of course, if \alpha = 100 well in any experiment of interest, we can ignore it, but arguably the default learning rate setting of Adam works reasonably well in most problems of interest too -but of course, we wouldn't call it "without learning rates" method. - Overall, this is an unusual and unconventional idea (which is great!), but it is frankly not presented in a clear way. I do not get a lot of intuition from the paper about _why_ this works, how it is similar/different from SGD, how the different quantities being updated (G, L, reward, \theta, w, etc) evolve over the course of a typical training run, etc. despite spending considerable time with the paper. I would suggest adding an additional section to the appendix, or something, which gives a lot more intuition about how and why this actually works. More specific comments: - Very high level: "backprop without learning rates" is a strange phrase. Backprop has no learning rate. It's an efficient algorithm for finding the gradients with respect to all parameters in your model with respect to the output. SGD has a learning rate. SGD is often used for training neural networks. In neural networks backprop is often used for finding the gradients necessary for SGD. But you don't need a learning rate to backprop; they are disjoint concepts. - In your introduction, it might be worth mentioning results that show that the learning rate is one of the most "important" hyperparameters, in the sense that if it's set wrong the model may not work at all, so its correct setting can have a strong effect on the outcome. - The relation of the second and third inequality in the proof after line 112 took me about 15 minutes to figure out/verify. It would be helpful if you broke this into a few steps. - Algorithm 1 is missing some input; e.g. it does not take "T" or "F" as input. - Calling \beta_{t, i} the "fraction to bet" is odd because it can be negative, e.g. if the gradients are consistently negative then \theta_{t, i} will be negative and 2\sigma(...) - 1 will be close to -1. So you are allowed to bet a negative amount? Further, it seems that in general 2\theta_{t, i} can be substantially smaller than G_{t, i} + L_i - I think you have redefined w_t. When defining coin betting you use w_t to refer to the bet at round t. In COCOB w_t are the model parameters, and the bet at round t is (I think) \beta_t, i (L_i + Reward_t, i). - I think most readers will be most familiar with (S)GD as an optimization workhorse. Further presentation of COCOB vs. variants of SGD would be helpful, e.g. drawing specific analogies between each step of COCOB and each step of some SGD-based optimizer. Or, perhaps showing the behavior/updates of COCOB vs. SGD on additional simple examples. - "Rather, it is big when we are far from the optimum and small when close to it." Not necessarily - for example, in your |x - 10| example, at the beginning you are quite far from the optimum but \sum_i g_i^2 is quite small. Really the only thing you can say about this term is that it grows with iteration t, I think. - In Algorithm 2, the loop is just "repeat", so there is no variable T defined anywhere. So you can't "Return w_T". I think you mean "return w_t". You also never increment, or initialize, t. - Setting L_{t, i} to the running max of the gradients seems like a bad idea in practice with neural networks (particularly for RNNs) because if at some iteration gradients explode in a transient manner (a common occurrence), then for all subsequent iterations the update will be very small due to the quadratic L_{t, i} term in the denominator of line 10 of algorithm 2. It seems like you would want to set an explicit limit as to the values of |g_{t, i}| you consider, e.g. setting L_{t, i} <- max(L_{t - 1, i}, min(|g_{t, i}|, 10)) or something. - What version of MNIST are you using that has 55k training samples? It technically has 60k training images, typically split into 50k for training and 10k for validation. - I would suggest using an off-the-shelf classifier for your MNIST experiments too, since you are missing some experiment details (how were the weights initialized, etc). - You say your convolutional kernels are of shape 5 x 5 x 3 and your pooling is 3 x 3 x 3, I think you mean 5 x 5 and 3 x 3 respectively. - What learning rates did you use for the different optimizers and did you do any kind of search over learning rate values? This is absolutely critical!